# Using Machine Learning Algorithms to Pool Data from Meta-Analysis for the Prediction of Countermovement Jump Improvement

**DOI:** 10.3390/ijerph20105881

**Published:** 2023-05-19

**Authors:** Indy Man Kit Ho, Anthony Weldon, Jason Tze Ho Yong, Candy Tze Tim Lam, Jaime Sampaio

**Affiliations:** 1Department of Sports and Recreation, Technological and Higher Education Institute of Hong Kong (THEi), Chai Wan, Hong Kong, China; 2The Asian Academy for Sports and Fitness Professionals, Chai Wan, Hongkong, China; 3Centre for Life and Sport Sciences, Birmingham City University, Birmingham B15 3TN, UK; 4Research Center in Sports Sciences, Health Sciences and Human Development, CIDESD, CreativeLab Research Community, 5000-801 Vila Real, Portugal

**Keywords:** artificial intelligence, sports science, sports medicine, research–practice gap, sports performance, strength and conditioning

## Abstract

To solve the research–practice gap and take one step forward toward using big data with real-world evidence, the present study aims to adopt a novel method using machine learning to pool findings from meta-analyses and predict the change of countermovement jump. The data were collected through a total of 124 individual studies included in 16 recent meta-analyses. The performance of four selected machine learning algorithms including support vector machine, random forest (RF) ensemble, light gradient boosted machine, and the neural network using multi-layer perceptron was compared. The RF yielded the highest accuracy (mean absolute error: 0.071 cm; R^2^: 0.985). Based on the feature importance calculated by the RF regressor, the baseline CMJ (“Pre-CMJ”) was the most impactful predictor, followed by age (“Age”), the total number of training sessions received (“Total number of training_session”), controlled or non-controlled conditions (“Control (no training)”), whether the training program included squat, lunge, deadlift, or hip thrust exercises (“Squat_Lunge_Deadlift_Hipthrust_True”, “Squat_Lunge_Deadlift_Hipthrust_False”), or “Plyometric (mixed fast/slow SSC)”, and whether the athlete was from an Asian pacific region including Australia (“Race_Asian or Australian”). By using multiple simulated virtual cases, the successful predictions of the CMJ improvement are shown, whereas the perceived benefits and limitations of using machine learning in a meta-analysis are discussed.

## 1. Introduction

Practitioners in sports science and sports medicine are encouraged to adopt evidence-informed practices, assuring safe and effective training and treatment methods. However, it is common to find research–practice gaps in these disciplines [1,2,3]. Reade et al. reported that 25% of Canadian high-performance coaches did not find the literature fit for their sport-specific needs, whereas 35% of coaches agreed that the research was irrelevant to their daily professional problems [4]. Meanwhile, over 50% of these coaches considered the research findings to be difficult to use. According to Keegan et al., practitioners in various disciplines encounter unstructured, messy, complex, and uncontrolled situations in the real world, and hence, they are required to frequently solve problems with poorly defined variables [2]. In general, research is often conducted under very confined conditions, which may be considered less relevant to real-world practice. In sports science, many studies were conducted in lab-based settings in highly controlled conditions, compared to strength and conditioning practices in the field with high dynamics, complexity, and interactions. Therefore, it is questionable whether the findings from the published literature are useful and applicable to real-life situations.

To bridge the research–practice gap, researchers from different disciplines proposed new and innovative solutions [1,5,6]. For example, Sandbakk et al. suggested that journals should publish more case studies from practitioners so that findings can be more comprehensible by coaches [1]. Meanwhile, Blonde et al. and Kim et al. proposed using real-world evidence to reflect actual clinical scenarios and practice to supplement traditional randomized controlled trials (RCT) and provide a more realistic and complete evidence-informed picture [5,6]. However, before getting real-world evidence-based studies to become more commonplace, various privacy and legal issues in different countries need to be overcome first. Moreover, such studies using unstructured real-world data with high complexity require extremely large datasets. Practitioners are required to adopt administrative workflows that accommodate the collection and analysis of big data [7].

Currently, most studies rely on small sample sizes, which are often not statistically powerful enough to draw definitive conclusions for making decisions. Therefore, meta-analysis procedures that pool data from multiple studies may be regarded as a potentially useful research method in sports science and medicine [8,9]. In addition to conventional pair-wise meta-analysis, network meta-analysis was proposed and adopted to further allow the comparison of the effects of different interventions to handle more complex real-world conditions [10]. Although these sophisticated research methods may allow researchers to produce more meaningful results, practitioners may not be able to use those findings to replicate or simulate the results on their athletes. One issue that practitioners may raise is the between-subjects heterogeneity and individual responses within studies [11]. This may be the case as data are commonly pooled and aggregated, only presenting mean values for a given population. Therefore, it is highly questionable whether the findings from the literature can accurately reflect and predict the potential effects or results in the real world using similar or mixed modalities on their athletes or patients with different backgrounds and conditions.

In this regard, the use of artificial intelligence techniques in performing predictive analytics can be a novel yet practical way to pool data from complicated datasets with multi-dimensional variables in producing algorithms for providing simulated results for real-world but unseen conditions. Nowadays, machine learning (ML), as one of the artificial intelligence (AI) methods, is widely used in various disciplines and industries for both predictive and prescriptive analytics in solving real-world problems on different scales, and therefore, it has also recently gained an increased importance in sports science domains. Several machine learning algorithms, such as random forest (RF) ensembles, support vector machine (SVM), gradient boosting model (GBM), and multi-layer perceptron regression (MLPR), used for neural networking in deep learning, can take advantage of pooling both categorical and numerical variables with a linear or non-linear relationship to produce models for predictive purposes. Although complicated machine learning techniques such as neural networks are usually regarded as “black boxes” and difficult to understand the mechanisms, the relevant importance or impact of each variable in the process of model generation is detectable in many machine learning algorithms [12,13,14,15].

Ideally, performing ML using big data from real cases to build predictive models can represent a broader population and reflect the true effect of mixed and complex interventions. However, this approach requires extremely expensive platforms and time-consuming primary data collection processes. Moreover, the quality assurance and monitoring of proper data entry, and the cleaning and processing of data from strength and conditioning practitioners in different places are difficult, whereas the meaningfulness and quality of the data greatly impact the results and ML performance. Therefore, predictive model development using pre-existing datasets (i.e., large database studies and secondary analyses) can also be a feasible approach [16]. Nevertheless, there is no single golden rule to assess the quality of the data from individual studies, whereas the meta-analysis is regarded as high-level evidence. Consequently, it is interesting and seems worth investigating whether the use of machine learning models, as powerful AI tools, can pool data from original individual studies used in meta-analyses to make accurate predictions on a target variable. 

To the best of our knowledge, this is the first study to adopt such a novel research idea, and therefore, we only chose the studies included from meta-analyses as the pilot. With the use of machine learning in the meta-analysis (MLMA) to predict the target variable, the results may be more definitive, practical, and understandable for practitioners to apply. Since biased conclusions from inaccurate findings in medical research may lead to serious injuries or lethal risks, instead of predicting the treatment effects for medical conditions, using performance-related variables to explore the strengths and limitations of this novel research idea can be safer and more feasible. For example, the countermovement jump has always been a key test for assessing power performance and monitoring neuromuscular status, and therefore, is extensively studied [17,18,19,20]. Although traditional meta-analyses have provided certain insights regarding the training effects and potential benefits, the findings and experimental procedures of the included studies were not always consistent. The mixed results or conflicting findings were not always fully explained, as the discrepancies observed in those studies may originate from general (population type, intervention duration, and intensity of the activity performed) and/or specific factors (e.g., jump height, peak power, relative peak power, relative power, mean power, peak velocity, peak force, mean force, rate of force development, eccentric time/concentric time, flight time/eccentric time, and flight time/contraction time using an unloaded and/or loaded CMJ) [20].

Therefore, the purpose of this study is to examine the accuracy of selected machine learning models in predicting the performance change of the countermovement jump (CMJ) from predictors available in interventional studies included in the selected meta-analyses. By pooling the data from a large amount of the literature included in the meta-analyses with diversified subjects, training backgrounds, programs, and modalities, it was expected to be able to generate accurate predictive models using machine learning techniques. After selecting the most appropriate ML algorithm, feature importance was analyzed to further understand the most relevant and impactful variables affecting CMJ performance. Lastly, by fitting several simulated case data into the model to make some real predictions, the current work can serve as predictive analytics to extrapolate the results to many unknown conditions using artificial intelligence. Finally, the characteristics of the predicted outcomes and potential limitations of this new research idea are discussed.

## 2. Methods

### 2.1. Source of Data

To include sufficiently large datasets, a thorough search of the literature was performed by one of the authors (JTHY) via Sport Discus, PubMed, and Google Scholar from 2007 to February 2020, with the following Boolean operators: (“Meta-analysis” OR “Meta-analyses” OR “Meta-Analysis” OR “Meta-Analyses” AND “Countermovement jump” OR “Counter Movement Jump” OR “Counter-movement Jump” AND published after Jan 2000)”. A total of 25 meta-analysis-related studies were initially identified. After the exclusion of non-training-related information in the title and abstract screening, 16 meta-analyses assessing changes in CMJ performance after a period of a systematic training program were found to fulfill the initial criteria [17,18,19,20,21,22,23,24,25,26,27,28,29,30,31,32]. Among these 16 selected meta-analyses, 15 followed a PRISMA protocol (Preferred Reporting Items for Systematic Reviews and Meta-Analyses) or performed a quality check using the Physiotherapy Evidence Database (PEDro) scale, Consolidated Standard of Reporting Trials (CONSORT), the examination of funnel plots, recommendations from International Society of Musculoskeletal and Neuronal Interactions (ISMNI) or guidelines specific for lower back pain, whereas only 1 meta-analysis did not report relevant items. After that, individual studies from selected meta-analyses were included in this study if they fulfilled the following selection criteria: (1) participants undertook an organized training program; (2) CMJ height was measured pre- and post-test as a dependent variable; (3) gender of subjects were specified and were not mixed-gender subject groups; (4) subjects involved were healthy and physically active or trained in a particular sport; and (5) the exact values for the CMJ height was presented for pre- and post-test. The 16 selected meta-analyses included a total of 406 individual papers. Studies that could not be accessed for full papers or those that did not meet the inclusion criteria were excluded. Consequently, 124 studies were finally deemed eligible for further machine learning modeling. Since no previous similar study provides clear inclusion and exclusion criteria for developing an accurate and successful model, this study uses a pragmatic method in which all relevant meta-analyses with the keywords matched and within the selected time frame are included for performing further ML analytics. Among several ML regression performance indicators, the coefficient of determination (R^2^) is more definitive in indicating how good the model is, whereas 0.8 can be used to clearly indicate a very good regression model performance [33]. If none of the ML models could achieve the R^2^ of ≥0.8 after retuning the hyper-parameters and reiterations of ML models, the inclusion criteria would be revised and loosened (e.g., expand the time frame) to increase the size of datasets for better ML performance.

### 2.2. Identification of Predictors

Before starting the model development, the candidate predictors potentially influencing CMJ performance were identified. Since an excessively large number of variables or subcategories within variables would substantially increase the model complexity, such as dimension and cardinality, the features were selected according to the literature support [34,35]. These variables included age, gender, ethnicity, sport, level of sport participation, the total number of training sessions, training methods, the use of periodization, the use of strength training movements (squat, lunge, deadlift, and hip thrust), the training volume, and the use of intraset rest (Focke et al., 2013; González-García et al., 2019; McCurdy et al., 2005; Mujika et al., 2018; Nigro and Bartolomei, 2020; Oliver et al., 2013; Perez-Gomez and Calbet, 2013; Rouis et al., 2014; Slimani, Paravlic, and Granacher, 2018) [26,36,37,38,39,40,41,42,43]. All variables and their definitions are presented in Table 1 while the supplementary information are presented in Table A1.

Due to inconsistent presentations of training intensities from various training methods such as “6 repetition maximum (RM)” in strength work, “85% of 1 RM” in weightlifting, or “bodyweight” in plyometrics, the “intensity” was discarded, but instead, the input of multiple training methods was allowed such that lower limb strength training represents training intensity with the use of at least 80% 1 RM in no less than two weeks of training. Since the type of sports background of subjects were diverse in the selected studies, they were summarized as “vertical based sports”, “horizontal based sports”, and “other sports” based on the characterized nature of sports movements to avoid an imbalanced dataset or cardinality issues [44]. Furthermore, training programs of intervention studies varied training volumes in different phases or periods. Instead of using the numerical format, volume was discretized from “volume per week” into a categorical format. A recent meta-analysis concluded that optimal CMJ improvements occurred when 3 sets per exercise, 5 repetitions per set, and 2 sessions per week were prescribed, while on average, 6 exercises per session were used in the studies. Therefore, a total of 180 repetitions per week was selected as the cut-off point, and those who adopted training with 180 repetitions or above were regarded as high volume [24].

As the selected studies had varied sample sizes, and original individual data for each subject were not accessible, to avoid the excessively small and imbalanced class size of features and to better reflect the weighted contribution from each study, an oversampling technique using synthetic data was performed. Rows of data were duplicated according to the sample size of a particular study. For example, for a study that presented data with a mean ± standard deviation of 10 subjects, 9 additional rows of identical data were synthesized (i.e., total of 10) to adjust proper weighting for machine learning modeling. Subsequently, a total of 3915 rows of data were included for further modeling.

### 2.3. Developing Machine Learning Models

Before building the models, one-hot encoding was applied to handle features containing multiple categorical values [45]. K-fold cross-validation was used by splitting the data into 5 independent sets of observations (K = 5) for each model development to avoid over-fitting and inflated results when the model complexity increased [46]. Eighty percent of data from the dataset were used to train the model, while 20% were unseen data for cross-validation. The dataset was further normalized using MinMaxScaler for providing enhanced model performance (i.e., data were scaled into the range (0, 1)) [47].

The dataset in this study was composed of both numerical and categorical variables; therefore, machine learning algorithms selected must be regressors capable of using both features to accurately predict numerical values. Since each machine learning algorithm may best perform under specific circumstances with a different number and format variables, multiple machine learning regression algorithms were employed including random forest (RF) ensemble, support vector machine (SVM), light gradient boosting method (LightGBM), and multi-layer perceptron regression (MLPR). This was to compare the model performance for selecting the most accurate model to produce the most sensible prediction of CMJ performance changes.

The RF, an ensemble method using a large number of decision trees to split the response based on binary nodes, was used as it was shown to be an accurate regression algorithm to predict numerical outcomes [14]. Meanwhile, SVM, another popular supervised machine learning method, is robust against over-fitting issues and able to fit linear regression functions into high-dimensional feature spaces to predict values with good accuracy [15]. The MLPR is a class of feedforward artificial neural networks that simulate the structure and operation of the human brain. It contains the input, hidden, and output layers, and permits the use of both numerical and nominal variables to produce classification or regression models. When compared to traditional multiple linear regression in solving complex tasks (e.g., climate prediction), MLPR was shown to produce higher accuracy [13]. The LightGBM is a recently enhanced method of the gradient boosting algorithm and is recommended for both high efficiency and accuracy [12]. All ML models were developed with the Scikit-Learn package in Python 3 using Jupyter Notebook.

### 2.4. Metrics and Model Tuning

The performance of the models was evaluated using mean absolute error (MAE), root mean square error (RMSE), and coefficient of determination (R^2^). Absolute percentage error (MAPE) was not used due to the frequent report of infinite or undefined values when errors are close to zero [6]. The formulae for the calculation of MAE, RMSE, and R^2^ are as follows:(1)$$RMSE=\sqrt{\frac{1}{N}\sum_{i=1}^{N}{\left(O_{i}-P_{i}\right)}^{2}}$$
(2)$$MAE=\frac{1}{N}\sum_{i=1}^{N}|(O_{i}-P_{i})|$$
(3)$$R^{2}=1-\frac{\sum {\left(O_{i}-P_{i}\right)}^{2}}{\sum {\left(O_{i}-P_{m}\right)}^{2}}$$
where *N* is the number of data points, while *O_i_* and *P_i_* are the observed and predicted values, respectively, and *P_m_* is the mean of *P_i_* values. The final training and testing performances in terms of MAE, RMSE, and R^2^ values were obtained by averaging the corresponding values from 5 sets of cross-validation. Training accuracy was obtained through the average values of MAE and RMSE of 5 sets of cross-validation. When the training accuracy of a model was lower (higher error values) than the validation one, it could be considered potentially underfitting, since an underfit model has not sufficiently learned the patterns of the training data well, and usually, the accuracy of both training and testing sets is low. In this case, further reiteration was performed after the retuning of hyper-parameters. Conversely, models were deemed to be overfitting if the validation accuracy was lower (higher error values) than the training accuracy. The model with the highest testing accuracy without underfitting and overfitting was regarded as optimum for further comparison and analyses [48]. A recent study highlighted that manual search could be more efficient to produce results with similar or even better model optimization than grid search, and therefore, the optimum prediction models of RF, MLPR, and LightGBM were obtained using random manual search with multiple reiterations after hyper-parameters were tuned [49]. Since each model development for SVM was time-consuming, hyper-parameter optimization was achieved using grid search.

For RF, the optimal model was achieved with the following parameters: max_depth = 50, n_estimatros = 500, max_leaf_nodes = 800, max_features = “sqrt”, bootstrap = False, and random_state = 0. Regarding SVM, the best model happened with the following: kernel = “poly”, C = 0. 1, epsilon = 1, and gamma = 0.1. For MLPR, the best model was yielded with the following: activation = “relu”, solver = “lbfgs”, learning_rate = “adaptive”, alpha = 0.0001, hidden_layer_sizes = (40, 80), learning_rate_init = 0.001, batch_size = “auto”, max_iter = 5000, and random_state = 0. In using LGBM, the model performed best with the following: num_leaves = 130, min_data_in_leaf = 3, max_depth = 15, learning_rate = 0.15, n_estimators = 200, lambda_l1 = 0.1, feature_fraction = 1, and xgboost_dart_mode = True.

## 3. Results

### 3.1. Prediction Accuracy

The model performance shown in Table 2 did not show any underfitting or overfitting problems, as both the MAE and RMSE of the training and testing sets were highly comparable in all models. Among the four selected machine learning algorithms, the RF model yielded the most accurate prediction with excellent testing accuracy (MAE: 0.071 and RMSE: 0.300), followed by the LightGBM, MLPR, and SVM models. Meanwhile, the RF and LightGBM models produced the highest R^2^ values (0.985). Therefore, the RF model was selected for further analysis due to the lowest error and highest R^2^ value.

### 3.2. Feature Importance

Figure 1 displays the feature importance score computed from RF for all 40 variables. The total number of features shown was expanded because of the use of one-hot encoding for the variables with multiple categorical values. The values of feature importance are the averaged impurity weighted in the trees of the forest. The top three features with the most predictive power in the RF model include the “Pre-CMJ”, “Age”, and “Total number of training_session”. These are followed by five other relatively important features including “Control (no training)”, “Squat_Lunge_Deadlift_Hipthrust_True”, “Squat_Lunge_Deadlift_Hipthrust_False”, “Plyometric (mixed fast/slow SSC)”, and “Race_Asian or Australian”. Interestingly, the features ranked ninth (“Level_junior team”) to twenty-eighth (“Level_Recreational/amateur/collegiate”) are similar and somewhat impactful on the prediction of CMJ improvement using the RF ensemble. However, the influence from the last two features, including the use of agility or quickness-related training programs (“Agility/Quickness training”) and other non-specific training methods not addressed explicitly in other features (“Other general resistance”), is almost negligible. The complete ranking and relative arbitrary score of feature importance are shown in Figure 1 as well.

## 4. Discussion

Predicting the performance change for athletes using certain strength and conditioning training programs is important to allow coaches to foresee the effects of their training programs based on the background of athletes and help avoid using sub-optimal programs. Currently, practitioners and athletes have to spend time and effort to realize the effectiveness of training programs through training monitoring, testing, and evaluation. When the performance of athletes plateaus, acute training variables or exercise prescriptions are further adjusted to optimize performance. Therefore, performance coaches are encouraged to use evidence-based training methods [4]. Nevertheless, due to experimental limitations, biased studies, individual responses, controlled experimental environments, and the heterogeneity of subjects within research studies, it is uncertain whether the observed outcomes can be applied in real-world settings [2,11]. Therefore, the current study explores the feasibility of a novel MLMA method to pool data used from a sufficiently large amount of the literature with diversified subjects and training methods to generate a more accurate prediction. The final production model using the RF ensemble yielded an extremely high accuracy with only 0.071 cm of MAE and low RMSE values compared to the changes in the predicted and actual CMJ performance. The novel concept of using MLMA to “intelligentize” pooled data to provide real-world performance predictions seems feasible.

The selected RF model in the present study identified the importance of a group of selected numerical and categorical features to predict the change in CMJ performance, while CMJ is consensually described as a key performance indicator [20]. Unfortunately, unlike simple transparent models such as decision tree or multiple linear regression, the underlying mechanism of the RF model in acquiring the relative feature importance for each predictor is somehow still regarded as a black box, which means people cannot fully interpret how the prediction is made [50]. Therefore, inputting data from simulated cases to observe the change in outcomes concerning the manipulation of important features may help us further understand the relationship between predictors and dependent variables.

The initial simulation referred to an 18-year-old male American recreational athlete without performing any training. All training-related variables were set to 0 or “False”, except the “Control (no training)”, “Volume_per_week_Low”, “Sport Other Sports”, Periodization_No periodized program used”, “Squat_Lunge_Deadlift_Hipthrust_False”, and “Intraset_rest_False” were set to 1, and the baseline CMJ value was set as 10 cm, and the predicted CMJ change was −0.19 cm, which was approximate to a 2% change. As shown in Table 3, without manipulating the other variables, by adjusting the most impactful predictor, “Pre-CMJ”, to 20, 30, 40, 50, 60, and 70 cm, the predicted CMJ changes were −0.24, −0.39, −0.02, −0.15, 0.31, and 0.30 cm, respectively. No further change was noticed when the value was either smaller than 10 or larger than 70 cm. Similarly, when the second most important predictor, “Age”, was changed from 18 to 10, 15, 20, 30, and 50 years, the predicted CMJ changes were 0.36, 0.10, 0.20, 0.28, and 0.27 cm, respectively. No further changes were observed when the “Age” was either smaller than 10 or larger than 50 years.

The RF model produced a non-linear prediction, whereas when producing the ensemble of decision trees, each one works like a bunch of if-else conditions. Therefore, the predicted outcomes from the RF model were not generated by mathematical formulas, but instead, only produced when certain conditions of decision trees were met. Based on the mechanism of the RF ensemble, it is possible that individuals with a particular age or baseline CMJ value can increase or decrease the CMJ performance under different circumstances when all other predictors are concurrently manipulated. Technically, the RF ensemble can enhance the model performance by building and aggregating multiple weak “learners” or “models” with specific strategies such as bootstrap aggregation or bagging. It is based on the parallel ensemble idea such that multiple base learners (decision trees) are constructed simultaneously and become a large forest. Therefore, the predicted values are within those seen in the training dataset, whereas this method is robust to noises or outliners. Conversely, traditional multiple linear regression allows the extrapolation of values that fall outside the training dataset. In this regard, machine learning using RF can protect users from obtaining nonsensical, extreme, or abnormal outcomes. However, it can also be a potential limitation that predictive RF models cannot be used to anticipate non-typical conditions such as excessively young or old individuals. It is also worth noting that in the current study, the LightGBM has shown highly comparable performance (with the same R^2^ and RMSE) to the RF as it is also one of the ensemble-based ML methods. The LightGBM is a gradient boosting decision tree method optimized for both high speed and accuracy. The base concept of LightGBM is one of the boosting methods for building sequential decision trees. Each subsequent tree is built to reduce the errors of the previous one. In contrast, RF focuses on the use of the bagging method in which deep trees are combined as a large forest. Therefore, theoretically, RF is robust in producing a model with a reduced variance, while the LightGBM focuses more on reducing bias. A recent study has shown a very similar result that both RF and LightGBM outperformed most other ML methods, while RF had a very slight advantage in the accuracy of the classification task [51].

When the initially simulated condition was further modified to another simulated subject who completed 4 weeks of lower limb strength training (i.e., “Control (no training)”) = 0 and “Lower Limb Strength” = 1), the predicted CMJ performance was greatly improved from −0.19 to 0.87 cm. Interestingly, the optimum CMJ improvement of 1.16 cm was yielded when the third most important feature, “Training_sessions”, was set to 16. Similarly, when the predictors ranked from fifth to tenth shown in Figure 1 were adjusted one by one while keeping the previously modified features, the simulated results were generated and are presented in Table 4. After adjusting the 10 most important features shown in Figure 1, the new simulated condition could be described as an Asian or Australian junior male athlete who has completed 16 sessions of periodized strength training including the squat/lunge/deadlift/hip thrust variants as well as plyometric exercises. When compared with the initial simulated subject without the training intervention, the CMJ performance change in the new condition would substantially increase by 4.69 cm.

In the present study, it was found that the baseline CMJ performance, age, and the number of training sessions completed were the top three most important predictors. Considering the diminishing return with the continuing increase in performance, it is commonly believed that individuals with lower baseline values should have a larger improvement window when compared with those highly trained athletes [52]. Nevertheless, multiple studies show that individuals with different training levels could have varied training responses including CMJ performance [53,54,55]. Moreover, it is worth noting that the baseline CMJ values can be potentially influenced by the level of athletes [17,27,31], genes and races [39], gender [56], as well as the nature of the sport in which the athlete participates [31,39]. Given such high complexity and interrelated factors between the baseline CMJ performance and the other variables, the non-linear results observed when we only tuned the “Pre-CMJ” feature are not surprising. Similarly, despite “Age” being the second most important feature, athletes of different ages may also have varied training responses. For example, maturation or puberty in adolescents and sarcopenia in aged individuals both potentially induce a certain impact on power and jumping improvement [57,58]. Furthermore, the effect of young or old age on CMJ performance could be masked or boosted when certain conditions are met. Therefore, unlike traditional studies investigating the impact of age in a particular controlled and unique circumstance, the impact of subject age on CMJ performance change could vary non-linearly and disproportionately in different simulated scenarios.

Since the machine (computer) was trained using a sufficiently large dataset based on the RF algorithm to “learn” and “understand” a massive amount of genuine and synthetic conditions produced from the selected literature, the predictions made theoretically reflected all possible scenarios within the “boundary” of the observations. As several studies highlighted the positive impact of using the squat, deadlift, hip thrust, and lunge exercise variants, plyometric exercises, and periodized training programs on power development [24,36,37,38,42,43], it is unsurprising that when switching on all these variables (set these predictors to 1 while the opposite predictors are set to 0), a substantial increase in CMJ performance was observed. Interestingly, an increase in CMJ improvement was identified when changing the race from “American” to “Asian or Australian”. Although it is well known that genetic factors contribute to athletic performance including power output [39,40], to the best knowledge of the authors, no previous study has directly compared the difference in CMJ improvement between Americans and Asians or Australians. Therefore, it is speculated that both the smaller sample size from the literature using Asian or Australian subjects, and coincidentally, better CMJ improvement observed in those studies contributed to such unexpected differences between the predictions.

Undoubtedly, the novel MLMA method using machine learning and the literature data to perform real-world predictions can potentially bridge the research–practice gap. Nevertheless, the present study using MLMA is not without limitations. The primary issue is the limited accessibility to a large amount of original raw data. Although the RF ensemble demonstrated extremely high model accuracy in our study, oversampling with synthetic data based on the sample size of individual studies was not equivalent to using true individual data points. The combination of using authentic and synthetic data for machine learning is somewhat similar to meta-analyses of aggregated data. We believed that such oversampling techniques would somewhat affect the final prediction and ranking of the feature importance. Moreover, using one-hot encoding substantially increased the dimensionality of the problem, whereas combining classes of certain categorical features and discretizing numerical predictors into categorical variables might lead to information loss, and subsequently affect the model performance. For instance, no clues were identified to understand and differentiate the differences between Asian and Australian individuals when these two classes were combined. Similarly, with a limited number of data points from different sports such as basketball, volleyball, soccer, rugby, and sprinting athletes, the individuals were re-categorized based on movement vectors. Given that the exercise intensity and volume were reported in the literature using different formats, while some programs included varied volume or non-linear patterns in different training phases, accurate and consistent data input for subsequent machine learning was almost impossible. Transforming the numerical training volume to either high or low with 180 repetitions per week as a cut-off point might unavoidably lead to a potential bias and neglect the impact of different training volumes on the dependent variable. The MLMA method used in our study was not a true typical meta-analysis, whereas all the data used for the modeling must have been used in the meta-analysis. Therefore, another limitation of our study is that some papers that failed to meet the inclusion criteria within these selected meta-analyses or those papers published before or after the given search timeframe may have been missed. Lastly, a standardized quality check was not completed for all selected papers; therefore, there is a possibility that low-quality meta-analyses, biased predictions, and false accuracies may have been included in this study.

## 5. Conclusions

The use of the RF ensemble machine learning for predicting CMJ performance change showed excellent accuracy and sensible prediction. By using a sufficiently large amount of data to train the selected model, the baseline CMJ performance, age, and the number of training sessions completed were found to be the top three most important features. Apart from the RF ensemble, LightGBM can also be a good ML alternative to produce accurate models for similar tasks. To better facilitate the use of machine learning for pooling data used in meta-analyses, full access to original raw data and a consistent presentation format of the variables in the individual literature are needed. It is proposed that future similar studies applying machine learning methods may also be used for predicting other performance indicators when large datasets are available with good transparency.

## Figures and Tables

**Figure 1 ijerph-20-05881-f001:**
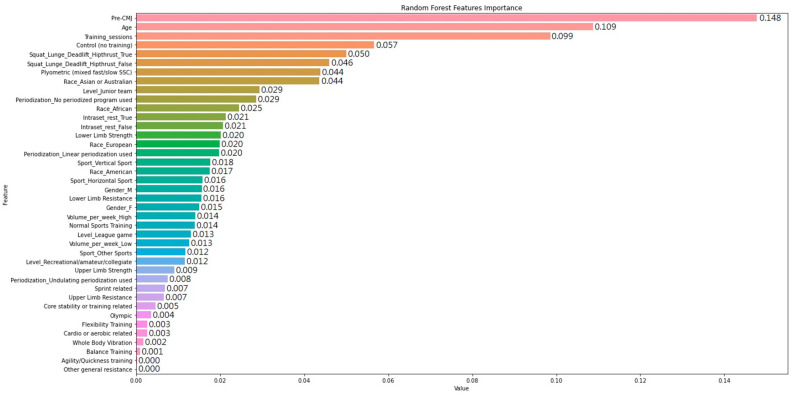
Feature importance of random forest on predicting improvement of countermovement jump.

**Table 1 ijerph-20-05881-t001:** Descriptions and formats of selected variables.

Variable	Description	Format
CMJ change	Raw difference of CMJ before and after intervention or control in cm by post-training CMJ–pre-training CMJ	Numerical
Age	Age of subjects	Numerical
Gender	Sex of subjects	Categorical
Ethnicity	Subjects’ ethnicity including American, Asian, Australian, African, and European	Categorical
Type of sport	Major movement directions or involvements in sports including horizontal based, vertical based, and other sports that are mixed or not well defined	Categorical
Level of sport participation	Level of subjects including league team, junior team, or recreational/amateur/collegiate	Categorical
Total training sessions	Total sessions completed by frequency × number of weeks. For the control group that received no intervention, 0 was inputted	Numerical
Method	Training methods or interventions involved in the study including control (no training), normal sports training, core stability or training-related, lower limb resistance, lower limb strength, plyometric (mixed fast/slow SSC), weightlifting, upper limb resistance, upper limb strength, flexibility training, balance training, sprint-related, agility/quickness training, cardio or aerobic-related, and whole-body vibration	Categorical
Special training drills	To indicate if training programs included squat, lunge, deadlift, or hip thrust movements or variants by marking true or false	Categorical
Periodization	To indicate if any periodization strategies were adopted within programs including no periodization, linear periodization, or undulating periodization	Categorical
Volume per week	To indicate the overall weekly training volume. Number of repetitions ≥ 180 per week indicated as high, while those <180 indicated as low	Categorical
Intraset rest	To indicate if intraset rest or cluster sets were adopted in the program by marking true or false	Categorical
Baseline CMJ	Indicated as the pre-CMJ value (cm) before intervention as the baseline	Numerical

**Table 2 ijerph-20-05881-t002:** Performance of RF, SVM, MLPR, and LightGBM models for prediction of CMJ improvement.

	RF		SVM		MLPR		LightGBM	
Metrics	Training	Testing	Training	Testing	Training	Testing	Training	Testing
MAE	0.069	0.071	1.18	1.20	0.134	0.141	0.072	0.074
RMSE	0.296	0.300	1.66	1.69	0.354	0.365	0.296	0.300
R^2^	0.985	0.985	0.523	0.510	0.978	0.977	0.985	0.985

**Table 3 ijerph-20-05881-t003:** The prediction of CMJ performance after changing “Pre-CMJ” value or “Age”.

Pre-CMJ Value (cm)	Age	Predicted CMJ Change (cm)
10 *	18 *	−0.19
20	18	−0.24
30	18	−0.39
40	18	−0.02
50	18	−0.15
60	18	0.31
70 or above	18	0.30
10	10	0.36
10	15	0.10
10	20	0.20
10	30	0.28
10	50 or above	0.27

* represents the baseline value for simulation.

**Table 4 ijerph-20-05881-t004:** The prediction of CMJ performance based on the particular simulated condition.

Variable	The Change Made	Predicted CMJ Change (cm)
Squat_Lunge_Deadlift_Hipthrust_True	1	1.61
Squat_Lunge_Deadlift_Hipthrust_False	0	2.08
Plyometric (mixed fast/slow SSC)	1	2.49
Race_Asian or Australian (meanwhile turn off “Race_American”)	1	4.04
Level_Junior team (meanwhile turn off “Level_Recreational/amateur/collegiate”)	1	4.36
Periodization_No periodized program used (meanwhile turn on “Periodization_Linear periodization used”)	0	4.69

## Data Availability

Not applicable.

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
