# Peer review of "Using Machine Learning Algorithms to Pool Data from Meta-Analysis for the Prediction of Countermovement Jump Improvement"

_ijerph, 2023, doi:10.3390/ijerph20105881_

Round 1
Reviewer 1 Report
This paper's goal is to fill a gap between research and practice in the area of high performance sport/coaching of high performance athletes through a meta analysis of previous literature where machine learning was used.
The manuscript contains many grammatical errors throughout. Below is not an exhaustive list but some of the mistakes have been noted.
Line 35 – grammar – many research studies were….
Line 36-37 – incorrect wording
Line 45 – last sentence of paragraph 2 is not a sentence.
Line 82-88 - sentence structure needs revising
Methods
Line 106: There are other relevant databases to search other than only Sport Discus, PubMed or Google Scholar (i.e. web of science)
I am not entirely clear why the authors would choose to look for only other published meta-analyses instead of just articles that have used machine learning techniques on indicator data for the counter movement jump.
Line 119: was this one study excluded? Please clarify inclusion and exclusion criteria.
Line 127: why would studies “not be accessed for full papers”?
Table 1: Please add row lines to delineate between items in the description, it is currently difficult to read.
Lines 211-216: can the authors provide a reference for similar methodology or justification for the methods used in this paragraph? In particular, the definition for overfitting and underfitting.
Table 1 discussion – it seems that the RF and LightGBM generate essentially the same MAE and RMSE and R2 values. The authors could comment on this similarity – and it seems they might have continued and compared the feature importance results from the two methods, which might be informative as well. At the very least, this second method deserves mention as well in the conclusions since it’s performance is at the same level as the RF.
I do not see Figure 1 in any of the submission materials to review.
Is “Pre-CMJ” the same as Baseline CMJ (table 1)? I don’t believe it is but I can’t find an explanation for this variable.
Table 3 – was does a negative CMJ change mean? I assume this means that the performance becomes worse. This table is not clear. Why would a preCMJ value of 10, 20, 40 and 50 result in worse performances (all other things being equal) but a preCMJ value is 30 result in a significant improvement – the largest improvement? I understand that the results of the model are non linear – but can the authors postulate a rationale?
Author Response
Dear Reviewer,
Thanks very much for the valuable and pinpointed opinions for enhancing our manuscript. We have made revisions according to your comments and here are our responses:
This paper's goal is to fill a gap between research and practice in the area of high performance sport/coaching of high performance athletes through a meta analysis of previous literature where machine learning was used.
The manuscript contains many grammatical errors throughout. Below is not an exhaustive list but some of the mistakes have been noted.
Thanks for pointing out this, we have made a substantial revision on the sentence structures and also the most obvious grammatical issues.
Line 35 – grammar – many research studies were….
This one has been corrected.
Line 36-37 – incorrect wording
This one has been revised as follows:
"In sports science, many studies were conducted in lab-based settings in highly controlled conditions compared to strength and conditioning practices in the field with high dynamics, complexity, and interactions. Therefore, it is questionable whether the findings from the published literature are useful and applicable to real-life situations."
Line 45 – last sentence of paragraph 2 is not a sentence.
This one has been revised as follows:
"Moreover, such studies using unstructured real-world data with high complexity require extremely large datasets. Practitioners are required to adopt administrative workflows that accommodate the collection and analysis of big data.[7]"
Line 82-88 - sentence structure needs revising
This one has been revised as follows:
"Although traditional meta-analyses have provided certain insights regarding the training effects and potential benefits, yet, the findings and experimental procedures of included studies were not always consistent. Mixed results or conflicting findings were not always fully explained, as the discrepancies observed in those studies may originate from general (population type, intervention duration, and intensity of the activity performed) and/or specific factors (e.g. jump height, peak power, relative peak power, relative power, mean power, peak velocity, peak force, mean force, rate of force development, eccentric time/concentric time, flight time/eccentric time, and flight time/contraction time using an unloaded and/or loaded CMJ).[20]"
Methods
Line 106: There are other relevant databases to search other than only Sport Discus, PubMed or Google Scholar (i.e. web of science)
I am not entirely clear why the authors would choose to look for only other published meta-analyses instead of just articles that have used machine learning techniques on indicator data for the counter movement jump.
Thank you for raising this issue. As the very first study using such a novel method with machine learning techniques, we simply used a pragmatic way that all focusing on all meta-analyses related studies to test the feasibility of using machine learning modeling. We admit that this method may not be the best and therefore we have also added relevant points in the section on limitations. Here are what we have added to supplement our manuscript:
Line 86-92:
"Nevertheless, there is no single golden rule to assess the quality of data of individual studies whereas the meta-analysis is regarded as high-level evidence. Consequently, it is interesting and seems worth investigating whether the use of machine learning models, as a powerful AI tool, can pool data from original individual studies used in meta-analyses to make accurate predictions on a target variable."
Line 150-154
"Since no previous similar study provides clear inclusion and exclusion criteria for developing an accurate and successful model, this study has used a pragmatic way that all relevant meta-analyses with the keywords matched and within the selected time frame were included for performing further ML analytics. "
Line 435-438:
"Therefore, another limitation of our study is that some papers that failed to meet the inclusion criteria within these selected meta-analyses or those papers published before or after the given search timeframe may have been missed. "
Line 119: was this one study excluded? Please clarify inclusion and exclusion criteria.
Thanks for pointing this out. We didn't exclude this. As the novel study, we simply used a pragmatic way to include all relevant meta-analyses. However, to make this study more replicable, we have also added that such a pragmatic selection method was checked using the ML regression performance indicator to give more concrete guidance on how to determine if we have used good and good enough data to produce a successful model for prediction purposes. Therefore, we have also added the following in the methodology section:
Line 150-159:
"Since no previous similar study provides clear inclusion and exclusion criteria for developing an accurate and successful model, this study has used a pragmatic way that all relevant meta-analyses with the keywords matched and within the selected time frame were included for performing further ML analytics. Among several ML regression performance indicators, the coefficient of determination (R2) is more definitive in telling how good the model is whereas, 0.8 can be used to clearly indicate a very good regression model performance.[33] If none of the ML models could achieve the R2 of ≥0.8 after retuning the hyper-parameters and reiterations of ML models, the inclusion criteria would be revised and loosened (e.g. expand the time frame) to increase the size of datasets for better ML performance."
Line 127: why would studies “not be accessed for full papers”?
Thanks for raising this. Basically, the full papers of all the searched meta-analyses were finally accessed. Only very few individual studies included in those meta-analyses were not accessible after sourcing all the databases in our workplaces, or the database from several universities. We have also sent a request to the authors but it was not accessible.
Table 1: Please add row lines to delineate between items in the description, it is currently difficult to read.
The table has been revised per the suggestion.
Lines 211-216: can the authors provide a reference for similar methodology or justification for the methods used in this paragraph? In particular, the definition for overfitting and underfitting.
We have added a reference [48] from another machine learning study using similar modeling methods. Since underfitting is a sign of insufficient use or learning of training data, therefore testing accuracy higher than training accuracy may be one of the hints in this regard. Therefore, we have also revised the sentence as follows:
Line 241-249:
"When the training accuracy of a model was lower (higher error values) than the validation one, it could be considered potentially underfitting since an underfit model has not sufficiently learned the patterns of the training data well and usually the accuracy of both training and testing sets is low. In this case, further reiteration was performed after the retuning of hyper-parameters. Conversely, models were deemed to be overfitting if the validation accuracy was lower (higher error values) than the training accuracy. The model with the highest testing accuracy without underfitting and overfitting was regarded as optimum for further comparison and analyses.[48]"
Table 1 discussion – it seems that the RF and LightGBM generate essentially the same MAE and RMSE and R2 values. The authors could comment on this similarity – and it seems they might have continued and compared the feature importance results from the two methods, which might be informative as well. At the very least, this second method deserves mention as well in the conclusions since it’s performance is at the same level as the RF.
Thanks very much for pointing this out. We agree that LightGBM has also done a very great job in this regard. Since the main theme of our manuscript was not to compare the performance of machine learning methods as a computer science / Ai oriented paper, meanwhile to keep the manuscript not to be excessively long, we keep focusing on the discussion on the top performing model only. However, we have added some discussion regarding the LightGBM per your suggestion as follows:
Line 351-361:
"It is also worth noting that in the current study, the LightGBM has shown highly comparable performance (with the same R2 and RMSE) to the RF as it is also one of the ensemble-based ML methods. The LightGBM is a Gradient Boosting Decision Tree method optimized for both high speed and accuracy. The base concept of LightGBM is one of the boosting methods for building sequential decision trees. Each subsequent tree is built to reduce the errors of the previous one. In contrast, RF focuses on the use of the bagging method which deep trees are combined as a large forest. Therefore, theoretically, RF is robust in producing a model with a reduced variance while the LightGBM more focuses on reducing bias. A recent study has shown a very similar result that both RF and LightGBM outperformed most other ML methods while RF had a very slight advantage in the accuracy of the classification task.[51]"
also Line 446-448 in conclusion:
"Apart from the RF ensemble, LightGBM can also be a good ML alternative to produce accurate models for similar tasks."
I do not see Figure 1 in any of the submission materials to review.
I hope you can see the file finally. We should have zipped all the figures based on the required submission format at the beginning. I also try attaching the figure file on this platform directly. Thanks.
Is “Pre-CMJ” the same as Baseline CMJ (table 1)? I don’t believe it is but I can’t find an explanation for this variable.
Yes. The Pre-CMJ is also the baseline CMJ. We have clarified this in table 1 now.
Table 3 – was does a negative CMJ change mean? I assume this means that the performance becomes worse. This table is not clear. Why would a preCMJ value of 10, 20, 40 and 50 result in worse performances (all other things being equal) but a preCMJ value is 30 result in a significant improvement – the largest improvement? I understand that the results of the model are non linear – but can the authors postulate a rationale?
Thanks so much for pointing out this. We have checked the table and paragraph back meanwhile re-iterated the model again to double-check. We found that the original paragraph stated -0.39 is correct while the 0.39 on the table has missed the -ve sign. Sorry for such a careless mistake. We believe sometimes such surprising or unexpected (non-linear) outcomes may exist as it also quite depends on what data we have given to the machine to learn. If we have most data from several papers only focusing on a particular age (e.g. 50 years old) and so happened all their control groups also improve for unknown reasons, the machine may also learn the pattern accordingly. Although the RF ensemble is already quite robust to handle noise or outliers, we can only safeguard a very sensible result when the dataset is sufficiently large to take care of multiple variables with high dimensions.

Reviewer 2 Report
Although the organizational structure of this article is correct, its research content is not fully explained in the article. I didn't seem to understand the intention of this article until I read the discussion part of the article.
Author Response
Dear reviewer,
Thanks very much for giving valuable comments and suggestions. We have made substantial revisions, especially the introduction section to make the research problem and purposes more explicit. Meanwhile, we have also revised many grammatical issues and sentence structure issues to enhance the readability further.
Line 24-123 of Introduction:
"Practitioners in sports science and sports medicine are encouraged to adopt evidence-informed practices, assuring safe and effective training, and treatment methods. However, it is common to find research-practice gaps in these disciplines.[1-3] Reade et al. reported that 25% of Canadian high-performance coaches did not find literature fit for their sport-specific needs, whereas 35% of coaches agreed that research was irrelevant to their daily professional problems.[4] Meanwhile, over 50% of these coaches considered research findings as difficult to use. According to Keegan et al., practitioners in various disciplines encounter unstructured, messy, complex, and uncontrolled situations in the real world and hence, they are required to frequently solve problems with poorly defined variables.[2] While in general, research is often conducted under very confined conditions, which may be considered less relevant to real-world practice. In sports science, many studies were conducted in lab-based settings in highly controlled conditions compared to strength and conditioning practices in the field with high dynamics, complexity, and interactions. Therefore, it is questionable whether the findings from the published literature are useful and applicable to real-life situations.
To bridge the research-practice gap, researchers from different disciplines have proposed new and innovative solutions.[1,5,6] For example, Sandbakk et al. suggested journals publish more case studies from practitioners so that findings can be more comprehensible by coaches.[1] Meanwhile, Blonde et al. and Kim et al. proposed using real-world evidence to reflect actual clinical scenarios and practice to supplement traditional randomized controlled trials (RCT) and provide a more realistic and complete evidence-informed picture.[5,6] However, before getting real-world evidence-based studies to become more commonplace, various privacy and legal issues in different countries need to be overcome first. Moreover, such studies using unstructured real-world data with high complexity require extremely large datasets. Practitioners are required to adopt administrative workflows that accommodate the collection and analysis of big data.[7]
Currently, most studies rely on small sample sizes, which are often not statistically powerful enough to draw definitive conclusions for making decisions. Therefore, meta-analysis procedures that pool data from multiple studies may be regarded as a potentially useful research method in sport science and medicine.[8,9] In addition to conventional pair-wise meta-analysis, network meta-analysis has been proposed and adopted to further allow comparing the effects of different interventions, to handle more complex real-world conditions.[10] Although these sophisticated research methods may allow researchers to produce more meaningful results, practitioners may not be able to use those findings to replicate or simulate the results on their athletes. One issue that practitioners may raise is the between-subjects heterogeneity and individual responses within studies.[11] This may be the case as data is commonly pooled and aggregated, only presenting mean values for a given population. Therefore, it is highly questionable if the findings from the literature can accurately reflect and predict the potential effects or results in the real world using similar or mixed modalities on their athletes or patients with different backgrounds and conditions.
In this regard, the use of artificial intelligence techniques in performing predictive analytics can be a novel yet practical way to pool data from complicated datasets with multi-dimensional variables in producing algorithms for giving simulated results for real-world but unseen conditions. Nowadays, machine learning (ML) as one of the artificial intelligence (AI) methods is widely used in various disciplines and industries for both predictive and prescriptive analytics in solving real-world problems on different scales and therefore, it has also gained increased importance in sports science domains recently. Several machine learning algorithms, such as random forest (RF) ensembles, support vector machine (SVM), gradient boosting model (GBM), and multi-layer perceptron regression (MLPR) used for neural networking in deep learning, can take advantage of pooling both categorical and numerical variables with linear or non-linear relationship to produce models for predictive purposes. Although complicated machine learning techniques such as neural networks are usually regarded as “black boxes” and difficult to understand the mechanisms, the relevant importance or impact of each variable in the process of model generation is detectable in many machine learning algorithms.[12-15]
Ideally, performing ML using big data from real cases to build predictive models can represent a broader population and reflect the true effect of mixed and complex interventions. However, this approach requires extremely expensive platforms and time-consuming primary data collection processes. Moreover, the quality assurance and monitoring of proper data entry, and the cleaning and processing of data from strength and conditioning practitioners in different places are difficult whereas the meaningfulness and quality of data greatly impact the results and ML performance. Therefore, predictive model development using pre-existing datasets (i.e., large database studies and secondary analyses) can also be a feasible approach.[16] Nevertheless, there is no single golden rule to assess the quality of data of individual studies whereas the meta-analysis is regarded as high-level evidence. Consequently, it is interesting and seems worth investigating whether the use of machine learning models, as a powerful AI tool, can pool data from original individual studies used in meta-analyses to make accurate predictions on a target variable.
To the best of our knowledge, this is the first study to adopt such a novel research idea and therefore, we have only chosen the studies included from meta-analyses as the pilot. With the use of machine learning in the meta-analysis (MLMA) to predict the target variable, results may be more definitive, practical, and understandable for practitioners to apply. Since biased conclusions from inaccurate findings in medical research may lead to serious injuries or lethal risks, instead of predicting the treatment effects for medical conditions, using performance-related variables to explore the strengths and limitations of this novel research idea can be safer and more feasible. For example, the countermovement jump has always been a key test for assessing power performance and monitoring neuromuscular status and, therefore, extensively studied.[17-20] Although traditional meta-analyses have provided certain insights regarding the training effects and potential benefits, yet, the findings and experimental procedures of included studies were not always consistent. Mixed results or conflicting findings were not always fully explained, as the discrepancies observed in those studies may originate from general (population type, intervention duration, and intensity of the activity performed) and/or specific factors (e.g. jump height, peak power, relative peak power, relative power, mean power, peak velocity, peak force, mean force, rate of force development, eccentric time/concentric time, flight time/eccentric time, and flight time/contraction time using an unloaded and/or loaded CMJ).[20]
Therefore, the purpose of this study was to examine the accuracy of selected machine learning models in predicting the performance change of countermovement jump (CMJ) from predictors available in interventional studies included in the selected meta-analyses. By pooling data from a large amount of literature included in meta-analyses with diversified subjects, training backgrounds, programs, and modalities, it was expected to be able to generate accurate predictive models using machine learning techniques. After selecting the most appropriate ML algorithm, feature importance was analyzed to further understand the most relevant and impactful variables affecting CMJ performance. Lastly, by fitting several simulated case data into the model to make some real predictions, the current work can serve as predictive analytics to extrapolate the results to many unknown conditions using artificial intelligence. Finally, the characteristics of predicted outcomes and potential limitations of this new research idea were discussed."
We have also added sentences/paragraphs in the discussion section to further enhance the technical discussion regarding the two top performing MLs.
Line 341-361:
"Technically, the RF ensemble can enhance model performance by building and aggregating multiple weak "learners” or “models” with specific strategies such as bootstrap aggregation or bagging. It is based on the parallel ensemble idea such that multiple base learners (decision trees) are constructed simultaneously and become a large forest. Therefore, predicted values are within those seen in the training data set whereas this method is robust to noises or outliners. Conversely, traditional multiple linear regression allows the extrapolation of values that fall outside the training data set. In this regard, machine learning using RF can protect users from getting nonsensical, extreme, or abnormal outcomes. However, it can also be a potential limitation that predictive RF models cannot be used to anticipate non-typical conditions such as excessively young or old individuals. It is also worth noting that in the current study, the LightGBM has shown highly comparable performance (with the same R2 and RMSE) to the RF as it is also one of the ensemble-based ML methods. The LightGBM is a Gradient Boosting Decision Tree method optimized for both high speed and accuracy. The base concept of LightGBM is one of the boosting methods for building sequential decision trees. Each subsequent tree is built to reduce the errors of the previous one. In contrast, RF focuses on the use of the bagging method which deep trees are combined as a large forest. Therefore, theoretically, RF is robust in producing a model with a reduced variance while the LightGBM more focuses on reducing bias. A recent study has shown a very similar result that both RF and LightGBM outperformed most other ML methods while RF had a very slight advantage in the accuracy of the classification task.[51]"
Reviewer 3 Report
This study examined the accuracy of four machine learning models (random forest ensemble, support vector machine, light gradient boosting method, and multi-layer perceptron regression) in predicting the performance change of countermovement jump using predictors pooled from the findings of meta-analyses studies.
The study is well designed, analyzed and presented. I have only one comment: More details about the information pooled from each study are needed. It would be helpful to included these in the appendix.
Author Response
Dear Reviewer,
Thanks much for the constructive comments given for enhancing the quality of this manuscript. We have made some amendments according to your suggestions. Here is our response marked as blue:
The study is well designed, analyzed and presented. I have only one comment: More details about the information pooled from each study are needed. It would be helpful to included these in the appendix.
Since traditional meta-analyses pooled and calculated the effect size from each included study, their focuses were more on what conditions or why such effects were produced. However, in machine learning-related research, we believe that the nature of data may give readers more insights into why certain ML algorithms may outperform others. Meanwhile, in this machine learning-related study, we have used a wide range of variables while some studies, for example, used very irregular training volume or frequency throughout their study.
To make this appendix more precise and insightful for understanding the training and testing dataset, we added a table per your suggestion and have only included the range (if numerical-based) and also the frequency (in terms of study groups if more categorical-based). We hope that such a table can on one hand provide additional but useful information meanwhile promote the readability of this paper.
Thanks very much.
Round 2
Reviewer 1 Report
The manuscript is much improved.
Reviewer 3 Report
All of my comments are addressed in this new version.